# Synthesis of Acrylate Dual-Tone Resists and the Effect of Their Molecular Weight on Lithography Performance and Mechanism: An Investigation

**DOI:** 10.3390/ma16062331

**Published:** 2023-03-14

**Authors:** Lifei Liu, Jintong Li, Ting Song, Rong Wu, Weizhen Zhao, Feng Huo

**Affiliations:** 1Beijing Key Laboratory of Ionic Liquids Clean Process, CAS Key Laboratory of Green Process and Engineering, Institute of Process Engineering, Chinese Academy of Sciences, Beijing 100190, China; 2School of Chemical Engineering, University of Chinese Academy of Sciences, Beijing 100049, China; 3Zhengzhou Institute of Emerging Industrial Technology, Zhengzhou 450000, China

**Keywords:** free radical polymerization, molecular weight, dual-tone resist, lithography, mechanism

## Abstract

Acrylate photoresists have gained considerable attention in recent years owing to their high resolution, high sensitivity, and versality. In this work, a series of thermally stable copolymers are synthesized by introducing an isobornyl group, and well characterized using Fourier transform infrared spectroscopy (FT-IR) and nuclear magnetic resonance spectra (^1^H-NMR). The effects of polymerization conditions on the molecular weight and their further influence on lithography are explored. By analyzing the thermal properties, film-forming capabilities, and the patterning behavior of these copolymers, a direct correlation between lithography performance and polymerization conditions is established via the molecular weight. In addition, the baking temperature of lithography is also optimized by atomic force microscopy (AFM), after which a line resolution of 0.1 μm is observed under the exposure of a 248 nm UV light and electron beam. Notably, our synthesized photoresist displays dual-tone resist characteristics when different developers are applied, and the reaction mechanism of acid-catalyzed hydrolysis is finally proposed by comparing the structural changes before and after exposure.

## 1. Introduction

Lithography technology has played a critical role in the semiconductor industry [1,2,3,4,5], enabling the miniaturization of devices and the reduction in semiconductor nodes from 10 mm to 7 nm over the past 50 years [6,7]. Photoresist is one of the key materials for nanofabrication in lithography, and the patterning generated by lithography acts as an etch-resistant agent to protect the silicon substrate. This allows the pattern on the mask to be successfully transferred to the silicon wafer, enabling the design of integrated circuits on silicon wafers. Therefore, the development of more advanced photoresists is particularly important for further achieving the goal of smaller nodes [8,9,10].

Typically, photoresists consist of a polymer backbone, photosensitive additives, and solvents, and the performance of the photoresist depends overwhelmingly on the polymer backbone [11,12]. Among the various polymer backbones used for semiconductor photoresists, acrylate copolymers have gained widespread popularity due to several advantages [13,14]. For instance, their transparent nature, especially for shorter wavelength light, allows them to be used universally for 248 nm, 193 nm, and electron beam lithography (EBL) [15,16]. Moreover, their tunable properties can be altered by introducing different groups, which can enhance image quality and resolution [17,18]. Numerous studies have been conducted on the lithography properties of acrylate copolymers in recent years. Wang et al. prepared P(SSNa-co-t-BMA) by free radical polymerization and investigated its photoacid generation efficiency and lithography performance under 248 nm exposure, obtaining a resolution of 0.35 μm [19]. Nandi et al. investigated the EBL lithographic performance of a GBLMA-MAMA-MAPDST copolymer resist, patterning 100 nm line/space features, which has a high sensitivity of 36.5 μC/cm^2^ [20]. Tejero et al. investigated the dual-tone behavior of P(HEMA-co-MAAEMA) copolymer resists and studied their optical properties and their potential for achieving grey (3D) lithography [21].

The lithography performance of different acrylate copolymers has been extensively studied, and the effect of molecular weight on their lithography properties has also been reported. As one of the most important parameters of polymers, the influence of molecular weight on its properties could not be ignored [22]. Polymers with different molecular weights may even exhibit large differences in properties [23,24,25]. Rathore et al. explored the effect of molecular weight on lithography performance using a PMMA photoresist [26]. It was found that for main-chain scission-type photoresists, the high M_w_ material has better performance than the low M_w_ PMMA. Patsis et al. investigated the effects of molecular weight and acid-diffusion on LER, mainly by means of simulations [27]. The simulations indicated that acid diffusion can be the major LER-modifying factor, and the effect M_W_ on LER was seen to be of secondary importance. However, most of the current studies have mainly examined the effect of molecular weight on lithography performance, with less research on the polymerization reaction and subsequent lithography overall.

In this work, new copolymers were synthesized using isobornyl methacrylate (IBMA) with a hexatomic ring structure instead of the conventional PHS monomer, copolymerized with methyl methacrylate (MMA) and hydroxyethyl methacrylate (HEMA). The chemical structure was analyzed using FT-IR and ^1^H-NMR, while the thermal stability of the copolymers was evaluated by a thermogravimetric analysis (TGA) and differential scanning calorimetry (DSC). The polymerization experiments were conducted at different temperatures, initiator dosages, and time to determine the relationship between polymerization conditions and molecular weights. On this basis, the photoresist was exposed under a 254 nm UV light and electron beam, and the effects of different molecular weights on the thermal stability, film forming, sensitivity, and patterning of photoresists were further investigated. Simultaneously, the synthesized photoresist was demonstrated as a dual-tone resist, and the acid-catalyzing hydrolysis mechanism of the lithography was also studied by FT-IR and ^1^H-NMR. The photoresist showed promising results, as it can be used not only for 248 nm and EBL, but it also has the potential for application in 193 nm of exposure, making it a more universal material.

## 2. Materials and Methods

Methyl methacrylate (MMA, 98%), isobornyl methacrylate (IBA, 98%), hydroxyethyl methacrylate (HEMA, 98%), dimethyl sulfoxide (DMSO, 99.8%), and 2, 2′-azobutyronitrile (AIBN, 99%) were purchased from Aladdin Reagents Company (Shanghai, China). Triphenylsulfonium chloride was purchased from Alfa Aesar Company (99.5%, Ward Hill, MA, USA). Tetrahydrofuran (THF), acetonitrile (CH_3_CN), and petroleum ether were supplied by Sinopharm Chemical Reagent Beijing Co., Ltd. (AR, Beijing, China), and there was no further purification before use.

Hydroxyethyl acrylate-based resists were prepared using free radical polymerization [28,29]. Typically, MMA, IBA, and HEMA were used as monomers and were dissolved in freshly distilled THF and CH_3_CN (2:1 *v*/*v*) along with AIBN (2–10 mol %). After three cycles of freeze-thaw vacuum degassing, the solution was placed on a magnetic stirrer for various hours at a certain temperature, and then precipitated drop by drop into a large amount of petroleum ether. The white powder obtained was repeatedly dissolved in tetrahydrofuran and precipitated with petroleum ether, and finally, dried in a vacuum drying oven overnight.

The FT-IR spectra of the copolymers were obtained using a Nicolet 380 instrument (Madison, WI, USA) over the range 4000–400 cm^−1^. The ^1^H-NMR spectra were collected using a Brucker ARX-400 Advance Spectrometer (Karlsruhe, Germany) in DMSO-*d6* as the solvent and tetramethylsilane (TMS) as the internal standard to identify the copolymer structure. The thermal properties were analyzed using DSC (Mettler-Toledo, Greenville, Switzerland) under a nitrogen atmosphere at a heating rate of 10 °C/min to determine the melting point, and DTG-60H (SHIMADZU, Tokyo, Japan) was used to analyze the thermal stability (*T*d, at 5% weight loss) with a rate of 10 °C/min under a nitrogen atmosphere. Gel permeation chromatography was performed with THF as the eluent to determine the molecular weights of the copolymers, which were calculated with respect to polystyrene as narrow M_w_ standards.

To prepare the resist, copolymers were dissolved in ethyl lactate to form a homogeneous solution and filtered through a 0.22 μm filter to remove larger particles. The solution was then spin-coated onto pre-cleaned silicon wafers (2 cm × 2 cm) using RCA cleaning at a rotation speed of 4000 rpm for 45 s. The coated wafers were prebaked at 110 °C for 60 s, followed by exposure to deep ultraviolet (DUV, λ~254 nm) at a dose of 1 mW/cm^2^, and electron beam lithography was performed using JEOL electron beam lithography system (JBX-6300FS, Tokyo, Japan) with a dose of 200 μC/cm^2^. The exposed thin film was then post-baked and developed. The surface morphology of the lithography pattern was characterized using atomic force microscopy (AFM, MultiMode 8, Karlsruhe, Germany) and scanning electron microscopy (SEM, SU8020, Hitachi, Japan).

The prepared photoresist solution was spin-coated onto transparent CaF_2_ wafer according to the procedure described above, and FT-IR was performed on the disc before and after exposure for investigating the structural changes that occurred. In addition, the photoresist solution before and after exposure was also characterized by ^1^H-NMR to further confirm the changes in its internal structure.

## 3. Results

### 3.1. Characterization of Copolymers

On the one hand, ^1^H-NMR and FT-IR are used to characterize the structure and composition of copolymers. The ^1^H-NMR spectra of monomers and copolymers are shown in Figure 1a. The proton of methyl (δ = 3.58 ppm) in MMA, the proton of hypomethyl (δ = 4.80 ppm) in IBA and the proton of methylene peak in HEMA (δ = 3.90 ppm) were found in the copolymers (labeled as A, B, C in the Figure 1a) and changed from sharp monomer peaks to typical broad peaks of polymer [20,21]. The composition of the copolymers was obtained by calculating the peak areas of corresponding characteristic peaks as 0.28:0.27:0.45. FT-IR analyses were further carried out to illustrate the structure of the copolymers in Figure 1b. The peak at 1735.6 cm^−1^ and 1199.5 cm^−1^ is the stretching vibration peak of C=O and the symmetric stretching vibration peak of C-O in MMA [30]. The peak at 1724.0 cm^−1^ and 1054.8 cm^−1^ is the stretching vibration peak of C=O and C–O in IBA. The peak at 3432.6 cm^−1^ and 659 cm^−1^ is the stretching vibration peak of -OH and out of plane bending vibration of -OH in HEMA. All the mentioned characteristic peaks appeared in the copolymers, while the disappearance of the C=C stretching vibration peak at 1024–943 cm^−1^ and the formation vibration peak of C=C at 939–910 cm^−1^ indicate that the copolymers were successfully synthesized. The above characterizations indicate that the free radical polymerization reaction occurred and copolymerization was successfully achieved. At the same time, the unreacted monomers were completely removed, and the copolymers maintained high purity without the presence of impurities.

### 3.2. Effect of Different Polymerization Conditions on Molecular Weight

The effects of different reactions such as temperature, initiator dosage, and reaction time on the molecular weight were investigated, and the results are presented in Figure 2. It was observed that the molecular weight of the copolymer increased gradually with an increase in temperature, reaching a peak of 42,000 at a polymerization temperature of 65 °C. Subsequently, the molecular weight of the copolymer tended to decrease when the temperature continued to increase [31]. This is due to the fact that the high temperature at the initial stage helps to increase the initiator activity, thus promoting the polymerization reaction. However, the reaction efficiency of the depolymerization and the side reaction is also significantly increased when the temperature is further increased, which leads to a decrease in the molecular weight instead [32]. The effect of initiator dosage on the molecular weight of the copolymers is shown in Figure 2b. With the increase in initiator dosage, the molecular weight of the copolymer gradually decreases, which is consistent with what is reported in the literature [33,34]. The reason behind this phenomenon is that a higher concentration of initiator in the system leads to a faster initial reaction rate, resulting in a larger number of polymerization fragments. This is because a larger number of fragments usually means a smaller molecular weight of each fragment, which eventually leads to an overall reduction in molecular weight. In contrast, the effect of time (as shown in Figure 2c) on the polymerization reaction is reversed. The molecular weight of the copolymer gradually increases with increasing time, determined by the nature of the chain reaction, which is also consistent with what has been reported in the literature [35].

### 3.3. Effect of Molecular Weight on Thermal Properties

Three copolymers with different molecular weights were selected, named polymer 2, polymer 5, and polymer 7, and their number-average molecular weights were 42,000, 36,000, and 22,000, respectively. Using the temperature at 5% mass loss as the *T*d, it can be calculated from Figure 3a that the thermal decomposition temperatures of polymers 2, 5, and 7 are 253.8 °C, 238.9 °C, and 180.0 °C, respectively. Obviously, the thermal decomposition temperature of copolymers is positively correlated with the molecular weight, and the higher the molecular weight, the higher the thermal decomposition temperature. In contrast, the effect of molecular weight on the glass transition temperature (*T*g) of the copolymer does not seem to be as significant. As shown in Figure 3b, the *T*g of polymer 7 and polymer 5 are 118.3 °C and 115.5 °C, respectively. When the molecular weight further increases to 42,000 (polymer 2), the *T*g increases to 118.4 °C accordingly. As far as we know, TGA is tested by measuring the weight difference caused by the loss of small molecule volatiles before and after heating [36]. For copolymers with a short chain, chain breakage tends to produce smaller molecules that are easier to volatilize, while large molecule copolymers may produce long chain fragments after bond breakage that do not necessarily cause volatilization. This may explain the phenomenon that the higher the molecular weight, the higher the thermal decomposition temperature. At the same time, it is well known that a higher molecular weight always means a larger molecular volume and intermolecular forces, which makes it necessary to overcome more resistance when deformation occurs, ultimately resulting in a higher *T*d or *T*g, as mentioned above.

### 3.4. Effect of Molecular Weight on Film-Forming and Lithography Properties

Subsequently, the spin-coating behavior and photosensitivity of copolymers with different molecular weights are investigated in Figure 4a. In general, the film thickness became thinner as the speed increased. When the photoresist was spin-coated at the same speed, the lower the molecular weight of the copolymer and the thinner the film obtained and vice versa, which also provides some guidance for the selection of film thickness in the future. In order to further investigate the effect of molecular weight on photosensitivity, the copolymers with different molecular weights were prepared into photoresists and exposed to a 254 nm UV light, and the changes of photoresist film thickness were measured at different exposure times. As can be seen in Figure 4a, polymer 7 with a smaller molecular weight exhibited a significant reduction in film thickness at the beginning of the exposure for 10 min, with a remaining film thickness of only 80%. After 60 min of exposure, the remaining film thickness was 0, i.e., the lithography was complete. For polymer 2 with a higher molecular weight, the reduction in film thickness was not obvious in the first 10 min of exposure, and only about 5% of the film was completely lithographed, leaving nearly 95% of the film. Until 60 min after exposure, about 15% of the film was still not fully exposed. Overall, the higher molecular weight copolymer has a slower sensitization curve, while the smaller molecular weight copolymers have a steeper sensitization curve. This indicates that the copolymers with a smaller molecular weight are able to respond to UV light faster, and thus, complete the photolithography process faster for the same exposure time.

Moreover, the lithography performance of the photoresist at different rotation speeds was also explored, as shown in Figure 4b. When the rotation speed was 2000 rpm, the obtained patterning was not uniform enough and there was obvious aggregation, which may be caused by the thick film. When the rotation speed was 3000 rpm and 4000 rpm, the pattern became clear and flat, especially at 4000 rpm. Finally, when the speed was 5000 rpm, the patterning was no longer clear, which means that the film may have been too thin to produce a blurred pattern. The figure shows that the film thickness of the photoresist is most suitable when the rotation speed is 4000 rpm.

### 3.5. Baking Temperature Optimization

To investigate the influence of baking temperature on lithography, the effects of soft-bake temperature and post-exposure temperature on lithography patterning were studied. The patterns on silicon wafer substrates were placed under the AFM for observation, and a cantilever with a needle tip moved up and down on the surface. The change in height at each point of the scan can be obtained from the change in force between the needle tip and patterns; thus, the morphology of the patterns can be further obtained. Figure 5a–d shows the effect of different soft-bake temperatures on the lithography patterns. When the soft-bake temperature is 90 °C (Figure 5a), the photolithography patterns are blurred and illegible, probably due to the large amount of solvent still remaining in the film when baked at a lower temperature. When the temperature increases to 110 °C (Figure 5b), the patterns become clear and easily recognizable, indicating that the solvent starts to evaporate at this temperature and the effect of the residual solvent on patterns is decreasing. As the temperature increases further to 130 °C (Figure 5c) and 150 °C (Figure 5d), the patterns become regular and show a similar morphology, indicating that these two temperatures are relatively suitable. Considering the thermal stability of the photoresist film, a lower temperature usually tends to be chosen as the soft-bake temperature. The effect of post-exposure bake temperature (PEB) on the lithography is shown in Figure 5e–h. When the PEB temperature was 80 °C and 100 °C, the exposed area showed narrower lines than the mask, probably because the lower temperature prevented the exposed area from completing the acid-catalyzed reaction. When the temperature was increased to 120 °C, lines were obtained with a good separation state and morphology. However, when the temperature was further increased to 140 °C, the pattern became indistinguishable from the other lines and blurred together, probably due to the dissolution of the patterns in the developer by the excessively high PEB temperature. In summary, when selecting the baking temperature, the thermal stability of the photoresist and the roughness of the lithography patterning should be taken into account, as well as the degree of the acid-catalyzed reaction.

### 3.6. Lithographic Patterning and It Dual-Tone Behavior

Based on baking temperature optimization, the baking conditions were chosen (soft bake under 130 °C for 60 s, PEB under 120 °C for 60 s). The prepared photoresist films were exposed to a 254 nm UV light for 30 min and further developed by IPA/MIBK (3:1). The SEM image of the developed patterning is shown in Figure 6a. After that, EBL (30 keV, 200 μC/cm^2^) studies of these resist-coated thin films were also performed under the same conditions, and the 300 nm and 100 nm line/space features under e-beam exposure can be observed by AFM in Figure 6b. When different photomasks were used, the lithographic patterning showed good capability with clear lines and obvious contrast between exposed and unexposed areas. Finally, a line resolution of 0.1 μm was achieved. In particular, we also observed that the resist exhibits dual-tone behavior when developed with a different developer, as shown in Figure 6c. When 2.38 wt% of tetramethylammonium hydroxide (TMAH) is used as a developer, the exposed area is dissolved by the developer and the unexposed area is retained, which exhibits the properties of a positive photoresist. When IPA/MIBK (3:1) is used as a developer, the exact opposite pattern appears. The exposed area is dissolved while the unexposed area is preserved as a pattern.

### 3.7. Structural Changes before and after Exposure

To further study the structural changes leading to the solubility switch, FT-IR and ^1^H-NMR studies were performed on the resist films before and after exposure. The FT-IR spectra of the photoresist films before and after exposure are shown in Figure 7a; the characteristic peak located at 1486 cm^−1^ disappears, which was attributed to the C–O vibrational peak between the isobornyl group and the oxygen atom. Meanwhile, a new –OH characteristic peak appeared at 1081 cm^−1^, indicating that the lithography process may involve the disappearance of the isobornyl group and the creation of a new –OH [37]. Based on this, the ^1^H-NMR of the copolymer before and after exposure was performed as follows. A solution of ethyl lactate dissolved with copolymer was directly exposed to a 254 nm UV light. This was then added drop by drop to IPA/MIBK (developer) and an immediate solid precipitate was obtained. The resulting precipitate was centrifuged, dried, and then dissolved in d-DMSO for an ^1^H-NMR analysis (Figure 7b). Obviously, the characteristic peaks of IBOM at 4.8 ppm (α-H of the isobornyl group) disappeared, indicating the disappearance of the isobornyl group. The characteristic peak of HEMA (3.93 ppm) and MMA (3.56 ppm) could still be found. At the same time, the characteristic peak of –COOH at 12.3 ppm appeared after exposure, which suggests the generation of –COOH during the lithography process [38]. All of these changes are consistent with FT-IR, as described above. According to the characterization, the mechanism of lithography must be related to the disappearance of isobornyl groups and the formation of substances containing –COOH. It has been reported that the isobornyl group falls off to form –COOH when it undergoes a hydrolysis reaction [28], which is consistent with our characterization above. At the same time, the generated –COOH makes the polymer insoluble in IPA/MIBK, which is also in accordance with our experimental results. Therefore, the mechanism of PAG catalyzing the hydrolysis of the isobornyl group was eventually proposed in Figure 7c.

## 4. Conclusions

In this study, acrylic copolymers were successfully synthesized and characterized by FT-IR, ^1^H-NMR, TGA, and DSC. The effects of polymerization reaction conditions (temperature, initiator dosage, and time) on the molecular weight were investigated. Synthesized copolymers with molecular weights of 42,000, 36,000, and 22,000 were selected for the preparation of photoresists, and the effects of different molecular weights on their properties were further compared. At the higher molecular weight of 42,000, the copolymers showed higher *T*d and *T*g of 253.8 °C and 118.4 °C, respectively. At the same time, copolymers with a smaller molecular weight could be fully exposed within 60 min and had better photosensitivity. In addition to this, a resolution of 0.1 μm could be achieved under the exposure of DUV light and EBL. Finally, the structural changes of photoresist films before and after exposure were characterized using FT-IR and ^1^H-NMR, and the photolithographic process of the acid-catalyzed hydrolysis of the isobornyl group was eventually proposed based on this. Further development of this dual-tone photoresist for ArF and EUV lithography is upcoming.

## Figures and Tables

**Figure 1 materials-16-02331-f001:**
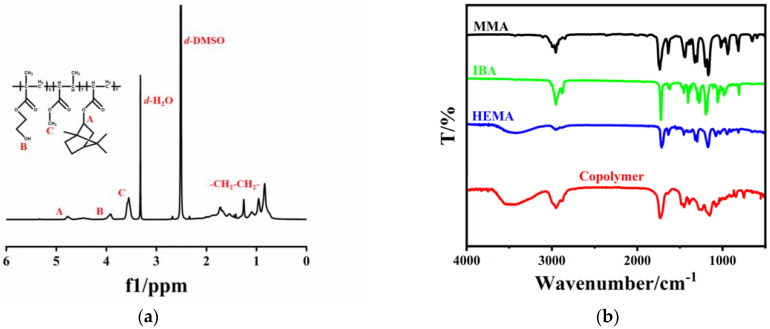
Structural characterization of copolymers (**a**) ^1^H-NMR spectra; (**b**) FT-IR spectra.

**Figure 2 materials-16-02331-f002:**
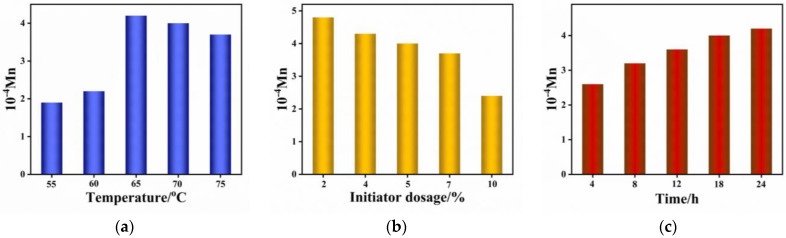
Effect of (**a**) temperature, (**b**) initiator dosage, (**c**) time on the molecular weight of copolmers.

**Figure 3 materials-16-02331-f003:**
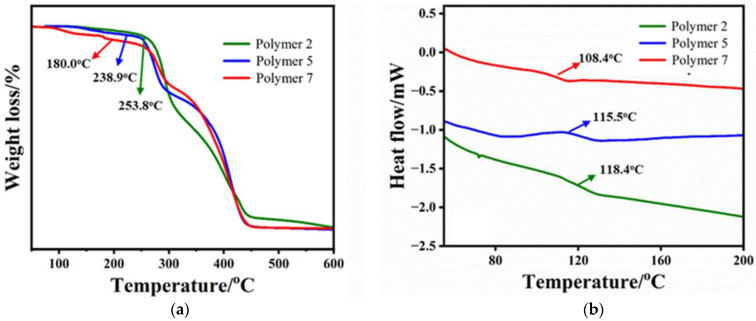
The effect of molecular weight on their thermal properties: (**a**) TGA curves; (**b**) DSC curve.

**Figure 4 materials-16-02331-f004:**
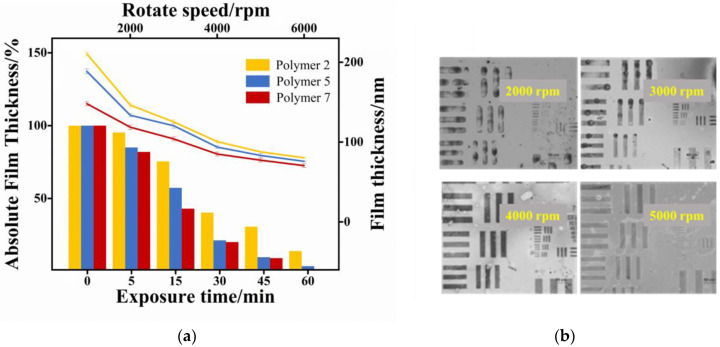
The effect of molecular weight on (**a**) film thickness at different rotation speeds and photosensitivity; (**b**) lithography patterning.

**Figure 5 materials-16-02331-f005:**
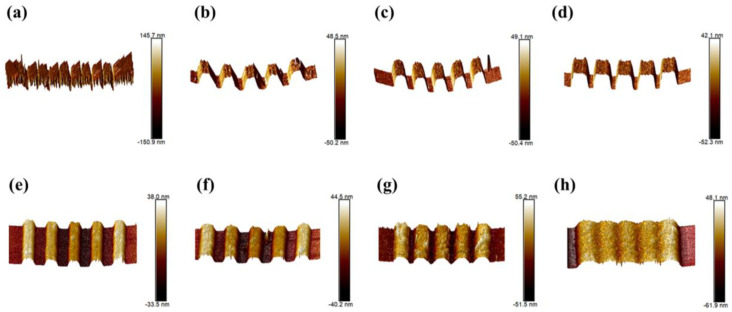
AFM images of lithographic patterning under different baking temperatures (**a**–**d**). Soft–bake: 90 °C, 110 °C, 130 °C, 150 °C; (**e**–**h**) Post–exposure bake: 80 °C, 100 °C, 120 °C, 140 °C.

**Figure 6 materials-16-02331-f006:**
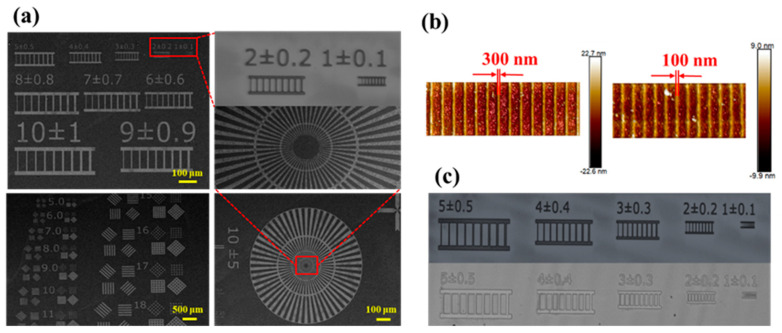
Lithographic patterning (**a**) exposed by different photomask; (**b**) EBL patterning; (**c**) developed by different developers.

**Figure 7 materials-16-02331-f007:**
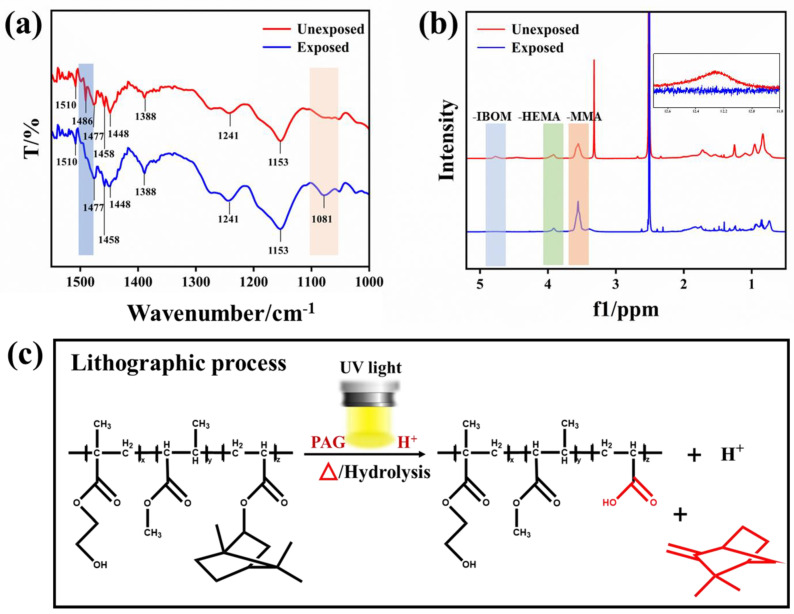
Structural changes before and after exposure: (**a**) FT-IR; (**b**) ^1^H-NMR spectra; and (**c**) the lithographic process.

## Data Availability

Not applicable.

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
