# Peer review of "Synthesis of Acrylate Dual-Tone Resists and the Effect of Their Molecular Weight on Lithography Performance and Mechanism: An Investigation"

_materials, 2023, doi:10.3390/ma16062331_

Round 1

Reviewer 1 Report

In this work Liu et al. synthesized acrylate based dual photoresist polymer and studied the effect of molecular weight on the lithographic performance. The authors mentioned that the effect of molecular weight of photoresist polymers has not been studied before. However, there are various studies on this topic (G. P. Patsis et al 2005 Jpn. J. Appl. Phys. 44 6341; Rathore et al 2020 j mat. Chem.C. 8, 5958, works by Christopher N. Anderson). The authors did not refer to these articles and did not establish how their work is different from the previous studies. The choice of monomers has not been discussed. Overall the introduction lacks impact of this work.

The data has not been well presented. For example, it is difficult to understand the FT-IR data if the peaks are not shown with markers or separately. It is also important to normalize each of the FT-IR spectra with respect to an internal peak that doesn’t change during the lithographic process. This is important to take care of the difference in overall intensity between two sets of experiments. The discussion on baking temperature and the corresponding AFM studies are ambiguous. In discussion the patterning and dual tone behavior the preparation conditions are not mentioned. It is important to run the FI-IR of the film after two different lithographic processes. In ascertain the mechanism it is important to run NMR on the film after dissolving it to solvent. It will tell which part of the polymer chains have undergone transformation.

Reviewer 2 Report

The article is interesting for researchers in area of copolymers, but must ammend and improve:

Which are the goals of the paper?

Which are the new findings and contributions of this paper?

Explain better lithography patterning in the paper and how can be studied by AFM, explain better this technique

In Which industrial applications can contribute your research? And how? 

Explain figure 7c--> How you can analyze the polymer structure? 

Explain better photolithographic process

In conclusions: Explain better and improve the explanatios of: the structural changes of photoresist films before and after exposure

Reviewer 3 Report

In the manuscript entitled "Synthesis of Acrylate Dual-tone Resists, the Effect of their Molecular Weight on Lithography Performance and Mechanism Investigation” authors synthesized a series of thermally stable copolymers by introducing isobornyl methacrylate (IBMA) with hexatomic ring structure instead of the conventional PHS monomer. They researched the effects of polymerization conditions on the molecular weight and their further influence on lithography performance. The polymerization processes were carried out at different temperature, initiator dosage and time, for obtaining the relationship between polymerization conditions and molecular weights.

The topic is interesting and relevant to the field due to the fact that research regarding the lithography technology is the foundation of integrated circuit and has greatly contributed to the development of the semiconductor industry. The chemical structure of copolymers was characterized using FT-IR and 1H-NMR and the thermal stability of the copolymers was tested by thermogravimetric analysis (TGA) and differential scanning calorimetry (DSC).

The manuscript is well organized, the authors used the scientific methods, and they are adequately described. The results are clearly presented and reproducible based on the details of the methods presented. In the conclusion part authors revealed the fact that dual acrylic copolymers have been successfully synthesized. They concluded that the property differences of copolymers with different molecular weight, such as thermal stability, film forming, photosensitivity and lithography patterning, were compared and accordingly presented.

The references cited in the manuscript are recent, mostly within the last 15 years. The figures, tables, images, and schematics are presented appropriately and clearly. Data presented in charts are properly presented and easy to interpret and understand.

The following corrections should be made:

- reference numbers in the text should be written before the period.

- conclusion chapter should be rewritten in a way to stress the main findings, with numerically presented main results.

Acceptance of the manuscript is suggested with those mentioned corrections.

Reviewer 4 Report

Dear Autor

It was very pleasant to receive and revise your manuscript. It is well organized, the grammar is excellent, the exposition of ideas and their discussion are very good, the conclusions are interesting and the references are updated.

I don't have repairs or revisions to make as I believe it's good as it is.

Thus, it is my pleasure to congratulate you for this nice work.

Round 2

Reviewer 1 Report

The authors have included the corrections as suggested. I suggest accepting the article in the present form

Reviewer 2 Report

Article has been ammended satisfactorily and can be published